# The balance stabilising benefit of social touch: Influence of an individual's age and the partner's relative body characteristics

Katrin Hanna Schulleri[1], Dongheui Lee[2,3], Leif Johannsen [4,5]*

1 Department of Human-centered Assistive Robotics, TUM School of Computation, Information and Technology, Munich, Germany, 2 Department of Autonomous Systems, Institute of Computer Technology, Technische Universität Wien, Vienna, Austria, 3 Institute of Robotics and Mechatronics, German Aerospace Center (DLR), Wessling, Germany, 4 Institute of Psychology, RWTH Aachen University, Aachen, Germany, 5 Department Health and Sport Sciences, TUM School of Medicine and Health, Munich, Germany

* leif.johannsen@psych.rwth-aachen.de

## Abstract

Interpersonal touch (IPT) is a successful strategy to support balance during a wide range of activities in daily life, including physical education and therapy. Despite common practice, however, the influence of individual characteristics – such as age, balancing skills, motor experience, sex and anthropometry – and differences between interaction partners on the balance stabilising benefit of social touch is unknown. In an opportunity sample of 72 pairs (age range 4–63 years), we assessed an individual's postural sway and change due to IPT during single-legged stance under four sensory conditions: with or without vision in combination with IPT or without. Following hierarchical cluster analysis (Ward's method) based on individual relative responses to IPT, individual and relative partner's characteristics two participant subgroups were identified: one of less stable, more vulnerable individuals, and another of more stable, mature participants. We developed multiple linear regression models, including moderating variables, to identify predictors of IPT benefit under each condition. Without visual input, an individual's benefit of IPT was determined by their balancing skill and the partner-related difference in balancing skill but not by any other factor or partner-related difference. Especially vulnerable individuals improved considerably with IPT when visual feedback was unavailable. When vision accompanied IPT, an individual's age-related motor developmental potential also played a significant moderating role. These findings indicate that the extent to which IPT is benefitting mutual balance stabilisation does not depend on biomechanical factors. Instead, the IPT benefit emerges as a product of both partners' sensorimotor capabilities and when visual feedback is available is also moderated by a person's motor developmental potential. We discuss a theoretical framework that accounts for the observed dependencies of the effect of haptic social support on balance control.

**Data availability statement:** All extracted data files are available from the figshare database (https://doi.org/10.6084/m9.figshare.27827127.v1).

**Funding:** This work was funded by the Department of Human Movement Science, and the Department of Human-centered Assistive Robotics of TUM, and the DFG SPP 2134 ("The Active Self") the Deutsche Forschungsgemeinschaft (DFG, German Research Foundation) - 402778716. The funders had no role in study design, data collection and analysis, decision to publish, or preparation of the manuscript.

**Competing interests:** The authors have declared that no competing interests exist.

## Introduction

Falls are a leading cause of injury-related deaths worldwide. One of the main causes of falls is impaired body balance [1]. Several factors are known to influence the stability of body balance, such as age [2–4], sex [5,6], and anthropometry [7–9]. Balance control can be facilitated by tactile feedback [10–13]. Just a light touch is sufficient to inform about any motion of one's own body relative to an earth-fixed reference point and thereby improve stability through optimised balance adjustments [14–16]. When physical contact is kept with an environmental reference, internal representations, such as the body schema, are involved in the localization of the contact and its relative motion in an egocentric frame of reference, taking into consideration the specific postures of the body and its limbs [17].

The stabilising benefit of external light touch generalises to social interactions. Interpersonal balance support is frequently observed in daily life, such as when providing support to a frail person in a clinical setting. Interpersonal touch (IPT) leads to enhanced balance stability, improved state estimation by augmented sensory feedback about own sway dynamics as well as tighter perception-action coupling minimising perceived fluctuations of the interaction forces may be mechanisms behind this effect. Passive exposure to the prerecorded sway dynamics of another individual via a haptic force feedback device, however, does not result in sway reductions in the way it is normally observed during contact with an actual human partner [18]. Therefore, sway reduction with IPT may reflect a mutually adaptive process between two contacting individuals and not just the reception of additional haptic information [19]. Furthermore, the effect of IPT does not seem to be the sole result of a mechanical coupling between both individuals but instead may represent the effect of mutually shared sensory information [20]. Thus, the social context during balancing with IPT seems to have an important influence in addition to possible (bio-)mechanical aspects.

During early motor development, haptic interactions with another individual play an important role for the development of postural control and the involved body representations and the ability to utilise touch may be fundamental for the development of a subjectively experienced self [21]. Bremner [22] characterised multimodal body representations as an interface between an individual's body and the external environment.. Early-stage toddlers when taking their first independent steps seem to be quite susceptible to haptic inputs that convey self-motion information gained from environmental contact 23]. Chen and colleagues interpreted this as an indication of progressively refined internal representations of own sway dynamics during standing and walking [23]. Similarly, Ivanenko et al. [24] demonstrated in toddlers how parental social touch improves postural stability in terms of reduced trunk sway, sideways hip motion as well as step width.

In children, the processing of sensory information, internal state estimation, and motor control are affected by greater amounts of internal noise [25–27] so that proprioceptive sensory information appears to have the greatest influence on balance stability compared to vision and tactile feedback [28]. Multisensory integration and

reweighting changes with age and is optimised in more mature young individuals [29–35]. Furthermore, internal representations and feed-forward, predictive control are developed gradually [36,37], while cognitive control continues to contribute to a greater extent than in adults [38]. Since children possess immature multisensory integration mechanisms and less precise body representations, they are less stable and may be more susceptible to haptic information compared to adults.

In old age, the developmental progression towards sensorimotor maturity observed in children and adolescents seems to be reversed. Noise in sensory feedback increases due to age-related deterioration in sensorial acuity [39], and sensory processing may slow down because of demyelination [40]. Therefore, older adults rely more on the integration of redundant information from multiple different sensory channels, such as vision [41,42] as the reliability of proprioception and vestibular sensation decreases [43]. Finally, as muscle strength weakens with age, potentially less optimal motor control strategies with a lower muscular effort are adopted [39]. All in all, one can argue that the motor developmental potential of older adults is minimal compared to children.

The developmental changes in childhood and adolescence and the deterioration in older age can be described as a U-shaped relationship [44] between age and balance skills, with a valley floor between 20–40 years (resembling an L-shaped function from 5 to 40 years) [3,45,46]. During childhood and adolescence, variability in balance performance decreases as stability increases [47] until the age of around 20–25 years, when a plateau is reached. After around the age of 45–55 years, body sway may begin to increase again, and efficiency of postural control deteriorates with older age [4,46].

A U-shaped relationship has also been observed between body sway and body mass index (BMI), which is a derived measure combining anthropometric characteristics such as body height and weight. Lee et al. [8] investigated the relationship between balance stability and BMI in a large cross-sectional study of community-dwelling older adults. An excessively increased BMI is associated with balance instability and increases an individual's risk of falling. Furthermore, individuals with extremely low BMI, for example due to eating disorders such as anorexia and bulimia, also showed reduced balance stability compared to individuals with normal BMI [48,49]. Interestingly, individuals with extreme BMI seem to demonstrate inadequate multisensory integration [50] and greater sensitivity to external tactile stimulation [51].

In addition to the state of sensorimotor development, ageing, and anthropometry, the gender or sex of an individual seems to have a distinct influence on balance control too [52]. Sensorimotor control and the integration of external feedback may be organised differently and specific moderating factors may be weighted differently between the genders. For example, studies on motion sickness [53] and 'mal de debarquement' syndrome [54] found females to be more susceptible to multisensory conflict. It has been hypothesised that central processes in multisensory integration, such as visual-vestibular integration, may differ between sexes (or gender identities) [55,56]. The influence of individual characteristics, such as age, anthropometry, and sex, on the benefit of IPT is unknown, and it is unclear to what extent also an interaction partner's individual characteristics play a role regarding the benefit of IPT. Consequently, the question arises if an "optimal" or most suitable IPT partner can be defined from whom one would benefit the most in terms of balance stabilisation. It is known that the less stable individual in a pair shows greater improvements in stability than the more stable partner. These effects of the tactile interpersonal interaction on balance stability could be mediated by biomechanical factors such as the weight, height, and the intrinsic stability of an individual and their partner effectively damping a less stable person's body sway. In the present study, we circumvented these biomechanical confounds by adopting an intrinsically more challenging standing posture, single-legged stance, where even a person with little body mass could easily perturb the balance of a person with a greater amount of inertia. Thus, we used this postural context to tease apart possible confounding factors determining the benefit of IPT on balance stability. In addition, performance in single-legged stance is strongly age-dependent [57].

We adopted a broad approach to investigate the potential factors influencing the benefit of IPT. We expected a greater benefit of IPT to be explained by an individual's balancing skill, by sex, by age (in terms of motor developmental potential), and by anthropometry (BMI). Furthermore, the benefit of IPT was also assumed to depend on the relative differences

in balancing skill compared to the partner's skill (i.e., greater benefit with a more stable partner), as well as in sex, age (greater benefit with a more mature partner) and BMI (greater benefit for individuals with extreme low or high BMI). Fig 1 summarises our conceptual framework of interactions between parameters.

## Methods

### Participants

Within the period from September 1st to November 30th, 2019, one hundred and sixty-two participants were recruited as an opportunity sample at public science festivals showcasing research undertaken at the Technical University of Munich. Individuals approaching our public display received an explanation of human sensorimotor control of body balance and the functioning of a barometric platform for stance and gait analysis and were invited in random pairs to take part in this study. Individuals in pairs could be related to each other (partners, parent and child, friends) or be unacquainted. All participants and their parental guardians, when participants were underaged, gave verbal informed consent. Capacity to consent was evaluated informally through the introductory verbal interaction in which a psychologist (L.J.) engaged participants in a conversation about research into people's balancing skills, including an explanation of the technical details of measuring body sway with a pressure distribution measuring plate, and the purpose of the current study on display. Individuals, who appeared incoherent during this interaction or unable to comprehend the explanations, were not offered an invitation for participation. Formal assessment of capacity to consent was not performed as the activity participants were required to perform (stand on one leg for a maximum duration 20 seconds) was of minimal risk and did not

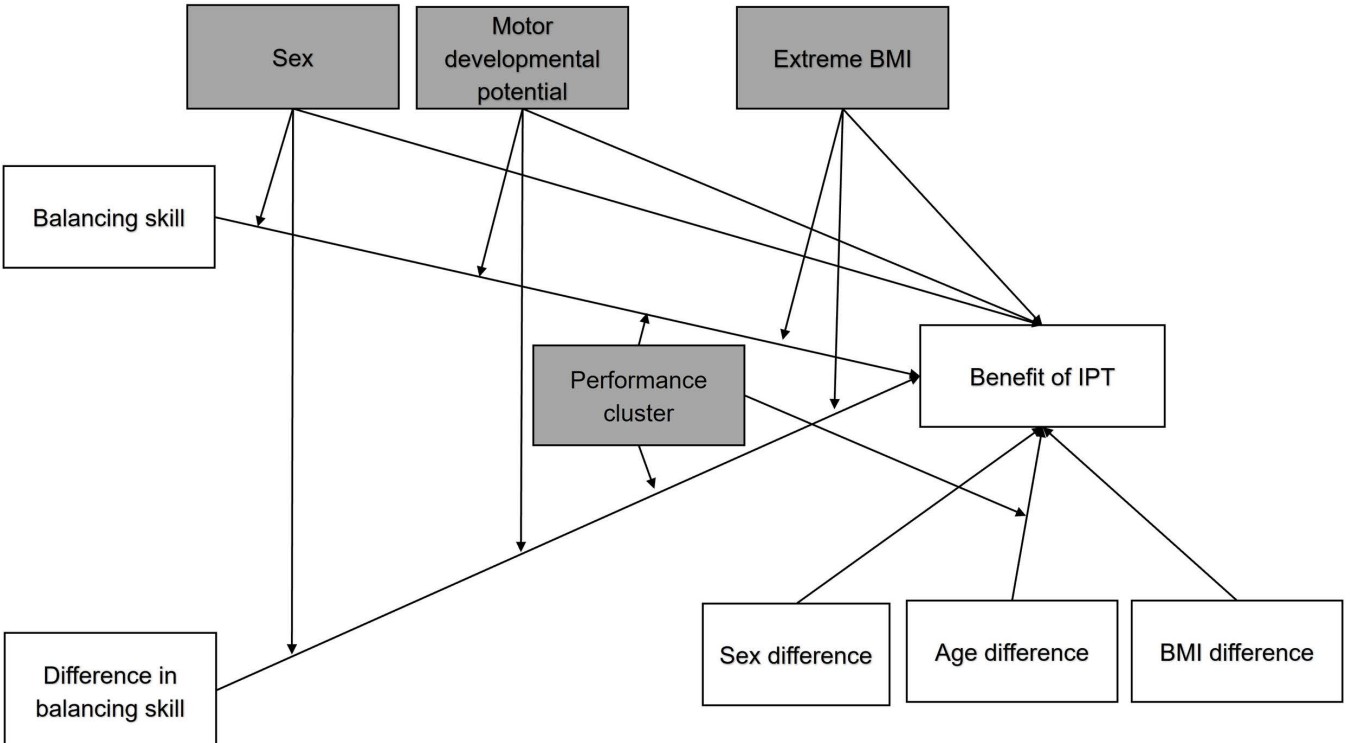

**Fig 1. Conceptual model of the influential factors on the benefit of IPT.** Sex, age-related motor developmental potential (mean-centred age inverse), extreme BMI (mean-centred BMI squared) and performance cluster assignment moderate the effects of an individual balancing skill, proportional interindividual differences in balancing skills, and interindividual differences in sex, motor developmental potential, and BMI. IPT: interpersonal touch; BMI: Body mass index.

involve any sensitive or invasive procedures. As the data collection was conducted in public the obtaining of consent was witnessed by any onlookers. Participants were excluded from subsequent data analysis, but not from taking part in the activity, if they reported pre-diagnosed sensorimotor impairments known to affect control of body balance, such as stroke or polyneuropathy. The experimental procedure was explained to each participant, and they could refuse to participate at any time. Participants remained anonymous throughout data collection and did not provide any personalised information that would make them identifiable. Only a participant's age, gender, body height and weight were recorded. Participants did not receive any evaluation or feedback about their balancing performance.

The investigation was carried out in accordance with the Declaration of Helsinki on Ethical Principles for Medical Research Involving Human Subjects and was approved by the medical ethical committee of the Technical University of Munich (2019–248-S-SR).

## Experimental design

The control of body balance can be challenged by a small base of support. Standing on one leg is a good indicator for an increased risk of falling [58] and is dependent on balance-specific feedforward and feedback sensorimotor control skills that are observed in experienced dancers [59]. Due to the relative difficulty and unstable nature of a single-legged stance (compared to a normal bipedal stance), we expected that also a taller and heavier individual's body sway could be destabilised severely by only slight perturbations imposed by an interaction partner. Therefore, we assumed that the body sway of shorter and lighter individuals would not be stabilised during IPT due to mechanical damping by a taller and heavier partner predominantly. Therefore, participants in a pair were instructed to stand side-by-side and orthogonal to the length of the pressure plate in a single-legged stance with stockinged feet on their preferred leg. Although data acquisition took place in public, participants had their backs turned towards any potential onlookers by facing a blank, white wall at a 2-metre distance. A pressure plate, two metres in length (Zebris FDM 2; single sensor dimensions: 8.46 mm * 8.75 mm; 240 * 64 sensors), was used to record the foot pressure distributions of a pair at a sample rate of 60 Hz. Four single trials of 20 seconds duration, one for each stance condition (Eyes open and Eyes closed both with and without IPT), were acquired so that the entire procedure lasted 5 minutes for each pair. The four conditions were tested in random order. When interpersonal touch was available, the individual on the left held the right hand in pronation to contact the fingers of the person on the right from above. Thus, the person on the right held their left hand in supination. For the administration of the interpersonal touch, participants in a pair were instructed not to alter their hand posture to grasp each other's hands but to rest their fingertips against each other, as reported by Johannsen et al. [13]. The balancing demands of single-legged standing, however, could result in participants deviating from their target posture, for example, when trying to rebalance themselves during a phase of instability, which could also result in variable forces. Participants were instructed to keep the same arm posture in all four trials and to remain relaxed without speaking during each trial.

## Data reduction

During post-processing, the pressure plate matrices for each data frame were divided into one area for each participant's footprint. From each trial, the longest period of static standing was visually determined and manually segmented. In the best of cases, the longest period of static standing covered the entire length of a trial. In those trials in which one of the two participants lowered their initially raised foot onto the plate, the longest uninterrupted period of single-legged stance was extracted. Subsequently, the Centre-of-Pressure (CoP) position was determined from the averaged pressure distribution during each data frame for each participant's foot. All data processing was conducted in MATLAB 2022b (Mathworks, Natwick, USA). CoP position time series were low-pass filtered at 10 Hz, and the displacement in each direction was used to calculate a frame-by-frame position change vector in the horizontal plane to yield a direction-unspecific rate of change measure of body sway (dCoP). Within-trial body sway variability was defined as the standard deviation of the rate of change measure (SD dCoP). Absolute and percentage changes in body sway variability resulting from the availability

of interpersonal contact were calculated. S1 Fig provides illustrative data traces of a pair of individuals standing with eyes closed with and without IPT.

## Statistical analysis

For the analysis of the influencing factors of sway change during interpersonal touch, we first excluded individuals as outliers with respect to their age, anthropometric parameters (height, weight, BMI), balancing skill in single-legged stance and relative interindividual differences in balancing skill. If an individual was excluded as an outlier, their paired partner was excluded too. We first computed a 2x2 repeated measures ANOVA to observe the commonly observed effects of vision and IPT on body sway. To investigate if the individuals' balancing skill (single-legged stance without IPT) showed an inverse or quadratic relationship with age [4,5] and a quadratic relationship with BMI, we computed a curved fitting analysis (S5 Fig). Curve fitting was performed based on age and BMI also for relative benefit of IPT (relative change in stability due to IPT) (S2 Fig).

To determine the number of personalised performance clusters present within the dataset, we performed a hierarchical cluster analysis based on individual responses to IPT (relative sway change), individual characteristics, as well as the partner's characteristics, (SPSS 28.0.1.0, IBM SPSS Statistics, Ward's method). A number of performance clusters had not been predefined. Participants' cluster assignments were then used as a moderator in the next step of backward bootstrapped regression analysis. The regression analysis served to predict the benefit of IPT (relative sway change) by the following predictors: an individual's balancing skill in single-legged stance (SD dCoP, no IPT), interindividual difference in balancing skill, sex, motor developmental potential (age mean-centred inverse; assumption of benefit of IPT approaching an asymptote in children), and extreme BMI (BMI mean-centred squared; assumption of benefit of IPT increasing in extreme BMI). Further, the relative differences between interaction partners in sex, age and BMI were included (Fig 1). We also included the interaction parameters, as we expected that the effect of an individual's balancing skill and interindividual balancing skill differences between partners on IPT benefit would be influenced by the performance cluster assignment, age, sex and BMI. For this, we multiplied an individual's balancing skill (SD dCoP, no IPT) and relative interindividual differences in balancing skills with the following moderators: motor developmental potential, BMI mean-centred squared, and the dummy coded performance cluster assignment and sex. In the next step, we computed a bootstrapped serial mediation analysis (N = 1000, 95%CI, seed 2021; PROCESS v4.0) [60] to further validate the expected relationships between age and anthropometric factors with individual balancing skill as well as between differences in age and anthropometric factors on the interindividual differences in balancing skill. The results of the relationship between age and anthropometric factors with an individual's balancing skill and with the benefit of IPT as well as detailed results of the mediation analysis can be found in the supporting materials. Bootstrapping was applied to augment data, as each condition consisted only of a single trial for every participant, as well as to counteract a potential non-normal distribution of the data. All statistical analyses were performed with SPSS 28.0.1.0 (IBM SPSS Statistics). The significance level was set to 0.05, and the statistical tendency level to 0.10. Cohen's d and partial $\eta 2$ are reported as effect sizes with low, moderate and strong effects defined as $d = 0.2$, $d = 0.5$, $d = 0.8$ and $\eta 2 = 0.01$, $\eta 2 = 0.06$ and $\eta 2 = 0.14$, respectively.

## Results

One hundred and forty-four individuals in the age range from 4 to 63 years (70 f, 74 m) were included in the analysis. Individuals were generally more stable with Eyes open and when IPT was available (Fig 2, S1 Table). Further, the benefit of IPT was greater in the Eyes closed (EC) condition (Mdiff = 75.73 (36%), p < 0.001, 95%CI [57.17 94.30]) than in the Eyes open (EO) condition (Mdiff = 13.58 (23%), p < 0.001, 95%CI [9.00 18.16]; F(1,143)=44.72, p < 0.001, $\eta 2 = 0.24$). The bootstrapped descriptive statistics of the whole group are further shown in S1 and S2 Tables.

The hierarchical cluster analysis, considering individuals' benefit of IPT (sway change due to the presence of IPT relative to standing without IPT), individual characteristics and relative interpersonal differences in characteristics (compared

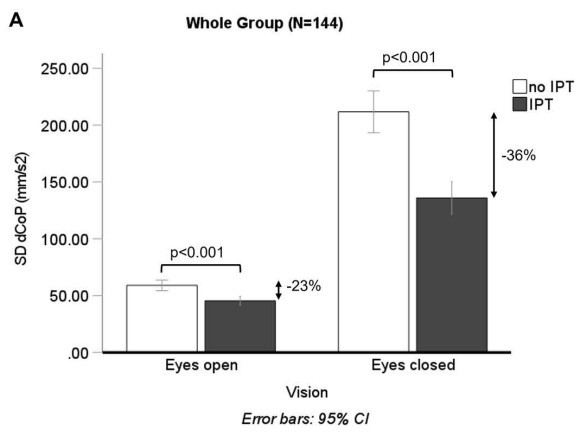

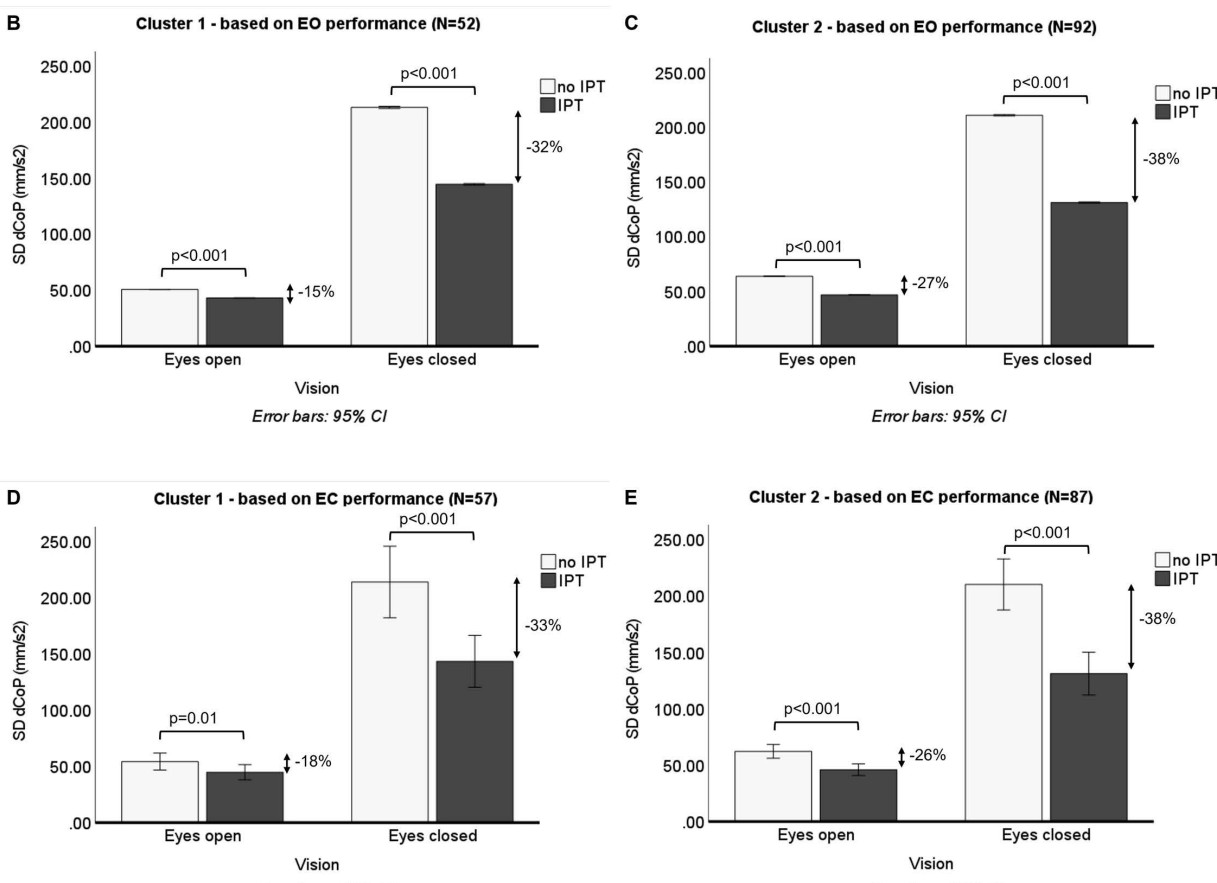

**Fig 2. Bar graphs of the effect of vision and interpersonal tactile interaction on variability of body sway velocity (marginal means).** Entire participants sample (A) and individually by performance cluster assignments based on behaviour during Eyes open condition (B, C) and Eyes closed condition (D, E). In general, a greater benefit of IPT is observable in the Eyes closed condition and for the participants assigned to Cluster 2 ('vulnerable' participants). Cluster 1: more skilled, older, taller and heavier participants; Cluster 2: more vulnerable participants. EO: Eyes open, EC: Eyes closed; IPT: interpersonal touch.

to the interaction partner), revealed two main clusters of participants (EO condition: Cluster 1: N = 52, Cluster 2: N = 92; EC condition: Cluster 1: N = 57, Cluster 2: N = 87). Participants' demographics and individual characteristics for the two clusters in the EO and EC conditions without IPT are shown in Tables S3 and S4, respectively. Comparing the two clusters, it becomes apparent that the first cluster included older, taller and heavier individuals, while the second cluster consisted of younger, smaller and lighter individuals, and consequently with a lower BMI.

Furthermore, in the EO condition, individuals in the second cluster demonstrated reduced single-legged balancing skill (in terms of greater variability of CoP velocity), relatively lower balancing skill within a pair, and a greater benefit of IPT (greater reduction of variability of CoP velocity due to IPT) compared to individuals in the first cluster. These differences between the clusters, however, were only observed in the EO, but not in the EC condition, which indicated that lack of vision also challenged the individuals in the first cluster considerably. In an additional bootstrapped bivariate Pearson correlation analysis (S1 Appendix; N = 1000, Seed = 2021, BCa95%CI) a stronger correlation was observed for an individual's balancing skill with the relative benefit of IPT (EO: r = −0.62, p < 0.001, BCa95%CI [−0.78–0.43]; EC: r = −0.69, p < 0.001, BCa95%CI [−0.72–0.61]) compared to the percentage benefit of IPT (EO: r = −0.36, p < 0.001, BCa95%CI [−0.52–0.22], EC: r = −0.42, p < 0.001, BCa95%CI [−0.50–0.34]).

Fig 3 depicts the relationship between interindividual differences in age-related motor experience and the benefit of IPT for each performance cluster separately. This figure further distinguishes between individuals whose partner was assigned

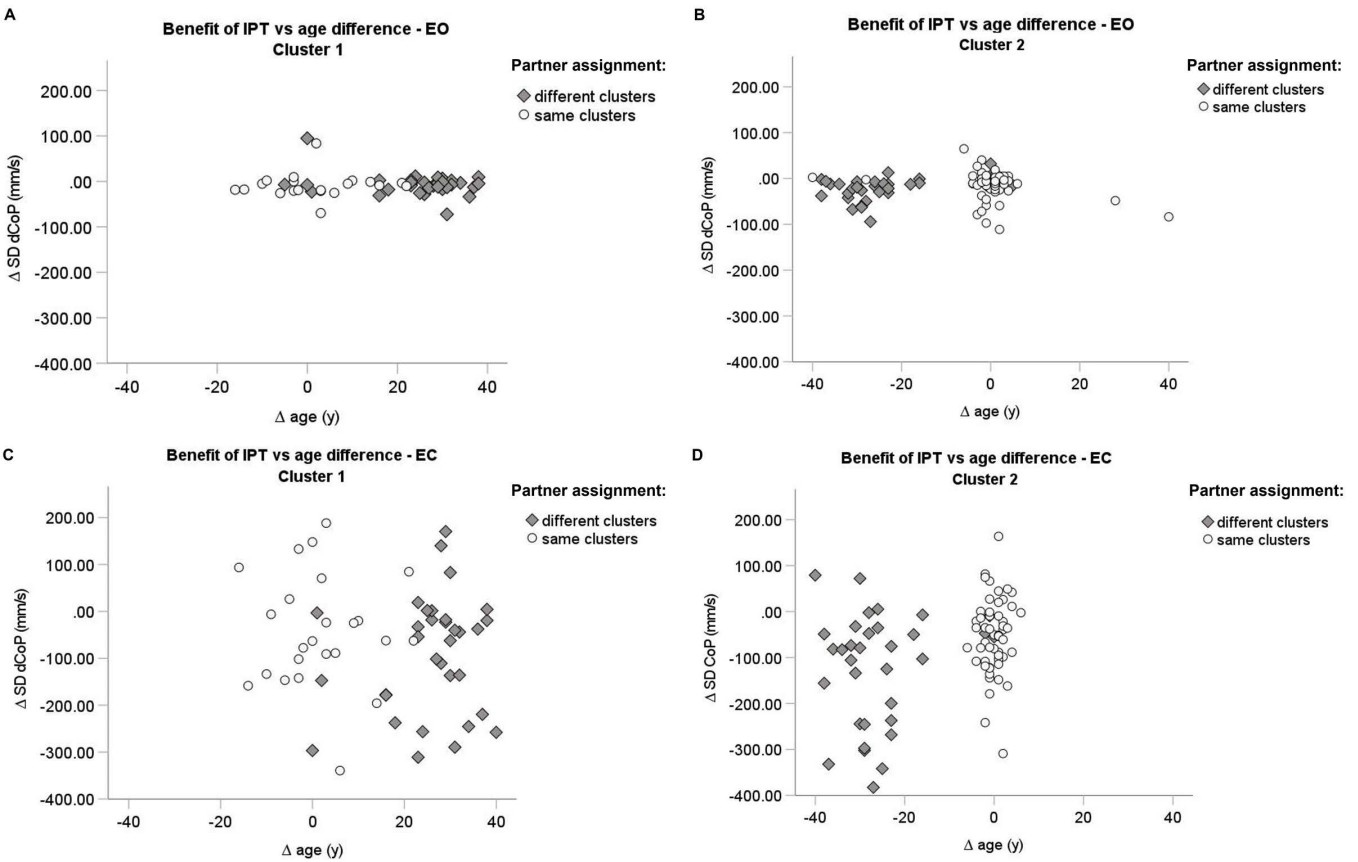

**Fig 3. Scatter plot of the benefit of IPT.** Eyes open condition (A, B) and Eyes closed condition (Fi. 3. C, D) by performance cluster assignment of an individual and performance cluster assignment of their interaction partner. Cluster 1: more skilled, older, taller and heavier participants; Cluster 2: more vulnerable participants. EO: Eyes open, EC: Eyes closed; IPT: interpersonal touch.

to the same or to a different cluster. For the EO condition (Figs 3A, 3B), differences in the benefit of IPT between the same and different partner cluster assignments were not observed, neither for the first nor for the second cluster (bootstrapped t-tests; Bias corrected and accelerated intervals: BCa; Cluster One: same cluster partner (N = 20): Mean = −8.58, BCa95%CI [−19.42 3.63], SD = 27.38, BCa95%CI [10.30 37.97]; different cluster partner (N = 32): Mean = −6.97, BCa95%CI [−14.18 1.35], SD = 24.58, BCa95%CI [11.74 32.88]; Mdiff = 1.62, p = 0.847, BCa95%CI [−13.24 15.47], Cohen's d = 0.06 95 CI [−13.24 15.47]; Cluster Two: same cluster partner (N = 60): Mean = −14.29, BCa95%CI [−22.03–6.85], SD = 30.30, BCa95%CI [22.77 37.05]; different cluster partner (N = 32): Mean = −21.99, BCa95%CI [−30.85–13.64], SD = 25.00, BCa95%CI [17.55 30.56], Mdiff = −7.70, p = 0.193, BCa95%CI [−18.73 2.82], Cohen's d = −0.27 95CI [−0.70 0.16]).

In contrast, in the EC condition, for the individuals in the second cluster a significant difference between individuals whose partner was assigned to the same vs. different cluster was observed. An individual with a partner in the first cluster showed a greater sway reduction (N = 33: Mean = −122.60, BCa95%CI [−166.18–76.14], SD = 122.72, BCa95%CI [98.49 139.67]) compared to individuals with a partner assigned to the same (second) cluster (N = 54: Mean = −52.54, BCa95%CI [−75.77–30.80], SD = 81.84, BCa95%CI [64.72 96.03]; Mdiff = −70.05, p = 0.007, Bca95%CI [−118.53–23.45], Cohen's d = −0.71 95 CI [−1.15–0.26]). However, comparing individuals in both clusters directly against each other as a function of the partner performance cluster assignments showed no differences between clusters, neither for partner assignment to the same nor to different clusters (Mdiff = 11.26, p = 0.670, BCa95%CI [−43.18 64.08], Cohen's d = 0,12 95 CI [−0.36 0.60]; Mdiff = 30.72, p = 0.335, BCa95%CI [−28.68 91.87], Cohen's d = 0.25 95 CI [−0.24 0.73]).

Bootstrapped regression analysis indicated that an individual's amount of relative sway change could be explained to 46% (p < 0.001) in the EO condition and to 52% (p < 0.001) in the EC condition (Fig 4A and 4B, respectively). Both conditions had in common, that individuals in the second performance cluster benefited more from an older, thus more experienced, interaction partner. (EO: B = 0.36, Bias = −0.03, SE = 0.11, p = 0.002, BCa95%CI [0.15 0.57], β = 0.17; EC: B = 1.44, Bias = 0.00, SE = 0.57, p = 0.013, Bca95%CI [0.25 2.54], β = 1.53).

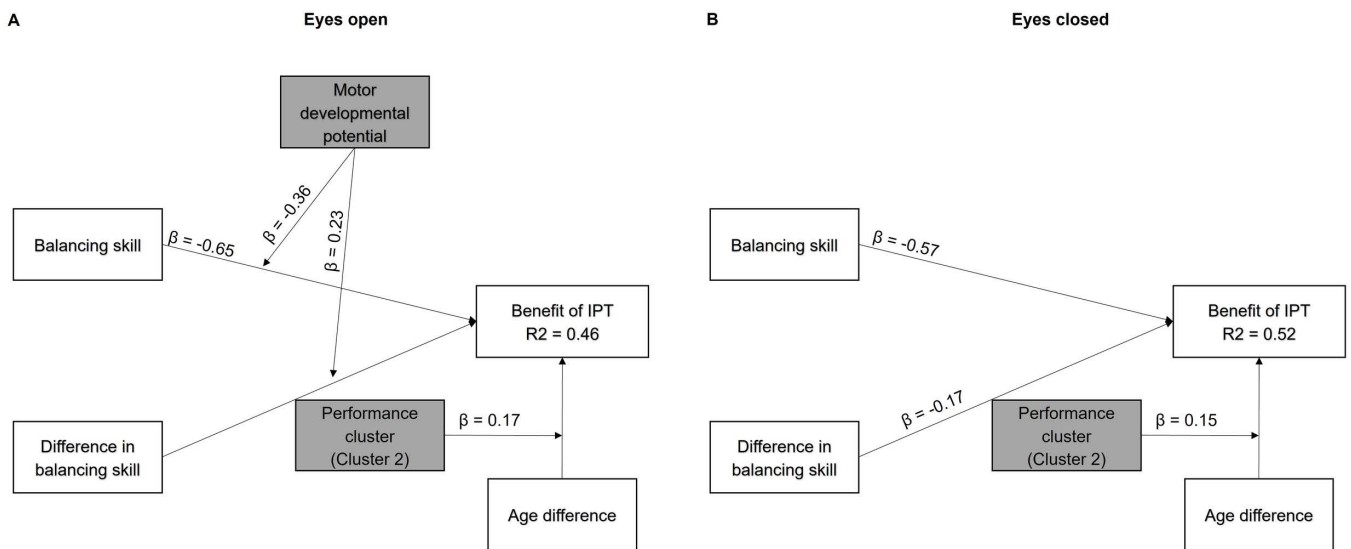

**Fig 4. Statistical model of significant influential factors on the benefit of IPT.** (A) In the Eyes open condition, the benefit of IPT depends on 1) an individual's balancing skill, 2) an interaction between the balancing skill by motor developmental potential, 3) an interaction between the interindividual differences in balancing skills and motor developmental potential, and 4) the age differences for individuals in Cluster 2. (B) In the Eyes closed condition, the benefit of IPT depends on 1) an individual's balancing skill and the 2) interindividual difference in balancing skills and 3) the age differences for individuals in Cluster 2. IPT: interpersonal touch, β: standardized coefficient.

Nevertheless, differences between regression models for the EO and EC conditions were also found. In both visual conditions, individuals with less balancing skill (greater single-legged sway variability) showed an increased benefit of IPT (greater relative reduction in sway variability; EO: B = −0.63, Bias = 0.06, SE = 0.09, p < 0.001, BCa95%CI [−0.80–0.42], β = −0.65; EC: B = −0.57, Bias = 0.03, SE = 0.09, p < 0.001, BCa95%CI [−0.74–0.42], β = −0.57). In the EO condition, however, the influence of balancing skill was greater in individuals with a greater developmental potential (younger age; B = −0.64, Bias = 0.04, SE = 0.32, p = 0.025, Bca95%CI [−1.09 0.03], β = −0.36).

Moreover, during EO, differences in balancing skill influenced more the benefit of IPT in individuals with a greater motor developmental potential, and thus, younger individuals (B = 43.65, Bias = −10.13, SE = 26.74, p = 0.082, BCa95%CI [1.49 67.40], β = 0.23). This means that the influence of interindividual differences in balancing skill as well as the influence of an individual's balancing skill on the benefit of IPT was enhanced with younger age (S4 Fig). In contrast, in the EC condition the influence of an individual's balancing skill and of interindividual differences in balancing skills was not moderated by an individual's motor developmental potential. In general, the benefit of IPT increased for less skilled individuals and for individuals paired with a more skilled interaction partner (B = −21.77, Bias = 0.51, SE = 8.70, p = 0.016, Bca95%CI [−38.03– 1.66], β = −0.17).

An additional serial regression analysis (S2 Appendix) showed that the variance of an individual's balancing skill could be explained by anthropometry and age to 14% and 18% only in both visual feedback conditions (S6 Fig).

## Discussion

Our study assessed the extent to which the stabilising benefit of the interpersonal touch (IPT) was determined by an individual's characteristics such as current age and potential in motor development (age inverse), height and weight of the body (via BMI), and an individual's balancing skill (in terms of body sway during single-legged stance without support) in combination with contextual factors such as the relative differences in the individual characteristics between interaction partners.

We confirmed that IPT improves stability of balance in single-legged stance (reduced variability in balancing performance) and that stance stabilisation is greater with IPT when visual feedback is not available. IPT resulted in sway reductions of 15–23% with and 32–38% without vision. This is a greater effect compared to previously observed reductions of 9–18% due to IPT in normal bipedal and Tandem-Romberg stances (greater reduction in Tandem-Romberg stance) [13,61]. The augmented reductions that we observed may be a consequence of the more unstable nature of a single-legged stance compared to previous studies with more stable stance postures. The amount of reduction in variability in balancing performance, especially without vision, is comparable to the effect of touching a static surface (20–31%) [62,63], when grasping another individual's shoulder (37%) [20], or when touching an artificial stabilising interaction partner (34–36%) [19].

Our findings indicate that the benefit of IPT is not driven by an individual's anthropometry or any interindividual differences in anthropometry. In our present study, instead, we found that an individual's stabilisation benefit of IPT was affected by an individual's balancing skill and by the relative differences between interaction partners' individual balancing skills as well as age differences. The amount of explained variance of balance performance by anthropometry that we observed was comparable to the explained variance reported in previous studies [3,4,7,64]. Thus, the influence of anthropometry on an individual's balancing skill and, thus, on the benefit of IPT are rather minor. This observation confirms a meta-analysis by Schmuckler [28], who found an anthropometric factor such as body height to influence balance stability in children only slightly. In the age range from 3 to 7 years body balance improved with increasing body height, possibly reflecting maturing sensorimotor control of balance, while at greater age body height imposed biomechanical constraints [2,65].

Irrespective of the visual feedback condition, IPT benefit was greater for more vulnerable, generally younger, smaller, and lighter individuals when interacting with a relatively older interaction partner. This was observed although these

participants were no longer in infancy or early childhood and therefore not reliant on any external, interpersonal locomotor support. At an average age of around 11 years, the more vulnerable participants of the second performance cluster supposedly had mastered their fundamental motor skills. Furthermore, as expected, a person's lower level of balancing skill was associated with greater balance improvements when IPT was available. This was observed independent of motor developmental potential when balance stability was challenged by the lack of visual feedback. On the other hand, when visual feedback was available, the individual balancing skill had a greater influence on the benefit of IPT for individuals with a greater developmental potential, and thus, younger individuals.

Remarkably, we also observed that individuals, who were relatively more unstable compared to their interaction partner during measurements without IPT, showed a greater benefit of IPT. This emphasises the importance of their balancing skill relative to their partner's balancing skill. This observation parallels previous reports where more unstable individuals standing in a Tandem-Romberg stance benefit more from IPT than their interaction partners standing in a more stable normal bipedal stance [61]. While this was observable independent of motor developmental potential when visual feedback was unavailable, in the condition with available visual feedback a relatively more stable interaction partner was more relevant for individuals with a greater motor developmental potential, and thus younger individuals.

The moderation of the effect of both balancing skills and differences in balancing skills on the benefit of IPT by an individual's motor developmental potential implies that the state of motor development of a participant (meaning balance control being refined less by experience) impacts the processing of visual information for balance control. In this situation, the more vulnerable individuals demonstrated comparatively high intra- (and inter-)individual variability in their level of balancing skill and the relative difference to their interaction partner (Fig 2, S3 and S4 Tables, S2, S3, and S4 Figs). In his meta-analysis, Schmuckler [28] rested the conclusion that the influence of proprioceptive information dominates balance control in children on the observed variable effects of stance width. This conclusion needs to be qualified by the fact that different stance postures not only entail an altered configuration of proprioceptive input but also confound differences in the intrinsic stability of a given stance posture (a wider stance is more stable in the mediolateral direction at least). Nevertheless, a single-legged stance as chosen in our present study generates quite salient muscle activations and proprioceptive feedback so that we do not see a contradiction with Schmuckler's conclusions [28].

The efficacy of balance control during a challenging single-legged stance may be limited by immature, less efficient multisensory integration, and less refined internal body representations in children and adolescents compared to adults [38]. Thus, a stronger influence of interpersonal differences in balancing skill during IPT may be associated with a greater susceptibility of vulnerable individuals to the haptic feedback received in terms of the interaction force [25]. The processing of proprioception for balance control remains underdeveloped until around 9 years of age [26]. Multisensory reweighting and integration continue to develop even in late childhood and adolescence [2,66,67]. Moreover, children have been shown to rely more on sensory feedback compared to adults [36,68] and they reweight and integrate inter-modal sensory information less adequately. For example, children are less capable to uncouple from tactile feedback at destabilising frequencies [25,69] and are more responsive to a wider range of tactile stimulus frequencies. Similarly, as in older age [42], noisier sensorimotor processing (measurement noise, estimator/computational noise, process/command noise), and a less accurate and precise, and thus more uncertain internal representations may require recalibration and refinement based on multisensory feedback during development. Thus, it is not surprising that these more vulnerable individuals showed greater intra- and interpersonal variability (S2 and S4 Figs) and a greater reliance on an older and more stable interaction partner than less vulnerable individuals and individuals with less motor developmental potential (Fig 3).

It is remarkable, however, that a similar moderating effect of motor developmental potential was not observed without visual feedback. In the condition without visual feedback, an individual's balancing skill and any interpersonal differences in balancing skills exerted a comparable influence on stability across the entire age range of the study. As standing on one leg without visual feedback is also challenging for adults, postural responses to IPT may also become more variable in adults (Fig 2, S4 Table, S2 and S4 Figs.). Nevertheless, the vulnerable individuals with a partner from the

other performance cluster showed greater sway reduction with IPT with eyes closed than those vulnerable individuals whose interaction partner was equally vulnerable (Fig 4). A similar difference in IPT-based sway reduction between non-vulnerable individuals with partners from the same cluster or partners from the other cluster was not observed.

The apparently greater dependency on an interaction partner with more dissimilar individual characteristics may indicate that more vulnerable participants, such as children and adolescents, possess less precise representations of their own body and its movement dynamics. Such limitations may become especially relevant when the visual channel does not contribute to feedback control of body balance. Uncertainty about the sensory consequences of own balance adjustments may make it harder to distinguish between self-induced sensory consequences and consequences of an interaction partner's balance adjustments. Therefore, when the relative differences to the characteristics of the interaction partner are more pronounced, a distinction between self- and other-induced sensory feedback may be facilitated, especially when the partner is more stable. Better distinction between self-evoked sway dynamics and dynamics evoked by the interaction partner might lead to more precise balance adjustments.

Moreover, when the representation of oneself is less precise and one's stance is unstable, individuals may be more likely to confuse consequences of their movements with the partner's. This confusion may lead to the commonly known interpersonal entrainment and resonance effects (e.g., synchronisation of each other's movements in the same direction). Lower intraindividual variability of a more mature interaction partner may increase the predictability of the partner [70] and therefore raise the likelihood of distinguishing the sources of sensory feedback for the more vulnerable partner during IPT. In contrast, if more mature individuals are affected less by sensory and control system noise and possess more efficient multisensory integration processes as well as more accurate and precise internal representations, they may be able to rely more on their own proprioceptive feedback and be less responsive to the haptic feedback granted by IPT. Their internal representations may allow a greater degree of predictive control when interacting with a more unstable or unreliable interaction partner, so any dependency on any partner's characteristics is diminished, which nevertheless does not preclude a beneficial utilisation of IPT.

Additional mechanisms involving social touch are also conceivable for the explanation of the benefit of IPT on body balance. For example, social touch has been demonstrated to facilitate reactions based on body-related mental representations [71]. Thus, IPT could have a direct effect on the perception of body motion, likely enhancing stability by improving both motor control and body awareness. Social distancing and social touch processing share neural bases with somatosensory region reactivity linked to social touch discomfort [72]. IPT may enhance balance by positively engaging somatosensory regions, reducing situational discomfort and subsequently improving sensorimotor integration.

Future studies are required to gain more insights into the role of own and relative partner characteristics on the benefit of IPT in the old (>80y) or oldest (>90y) population. As older aged individuals are known to rely more on concurrent delayed and more noisy multisensory feedback for continuously updating internal models, we hypothesise to find similar effects in older age; however, in the opposite direction, meaning that older individuals are expected to benefit more from a relatively younger interaction partner. Further, IPT in a balancing task with a more skilled partner may be used therapeutically to improve physical and psychological well-being in individuals with reduced mental health due to major depression [73].

## Conclusions

This study showed that anthropometric parameters, such as body height and weight, only play a minor role for the benefit of interpersonal touch, and that the developmental motor experience as well the single-legged balancing skill play major roles. Besides an individual's characteristics, such as balancing skill and age-related motor experience, also the relative differences in characteristics to an interaction partner contributed to the benefit of interpersonal touch on balance stability. We found that especially children and individuals with reduced balancing skill benefited more from interpersonal touch and that these individuals also benefited more from interacting with a relatively older partner. IPT is a balance support strategy that seems to benefit stability in a broad range of individuals, especially vulnerable individuals interacting with more stable

partners, and in challenging stance contexts, where mutual destabilization is likely. Thus, IPT is a promising approach in clinical contexts, such as routine activities and training sessions.

## Supporting information

**S1 Table. Descriptive statistics of the entire participant sample: individual characteristics.** EO: Eyes open, EC: Eyes closed; IPT: interpersonal touch.
(DOCX)

**S2 Table. Descriptive statistics of the entire participant sample: relative interindividual differences in characteristics between interaction partners.** EO: Eyes open, EC: Eyes closed; IPT: interpersonal touch.
(DOCX)

**S3 Table. Participant characteristics for the two performance clusters based on the Eyes open condition.** Significant cluster differences are indicated by a star and statistics are shown in the last column (equal variances not assumed). EO: Eyes open, EC: Eyes closed; IPT: interpersonal touch.
(DOCX)

**S4 Table. Participant characteristics for the two performance clusters based on Eyes closed condition.** Significant cluster differences are indicated by a star and statistics are shown in the last column (equal variances not assumed). EO: Eyes open, EC: Eyes closed; IPT: interpersonal touch.
(DOCX)

**S1 Fig** Illustrative data traces showing two individuals during side-by-side single-legged stance on a barometric platform. with (A) Eyes closed without interpersonal touch and with (B) Eyes closed and simultaneous interpersonal touch. The participant on the left of the pair is shown as a line in Blue, the participant on the right as a line in Orange. The 2D displacement vector magnitude is the resultant of the anteroposterior and mediolateral positions. IPT: interpersonal touch.
(TIF)

**S2 Fig** Curve fitting results for age (left) and BMI (right) with the benefit of IPT in Eyes open condition (EO; left) and Eyes closed condition (EC; right).
(TIF)

**S3 Fig** Scatterplot of the benefit of IPT (y-axis) in relation to an individual's balancing skill (x-axis). A more negative delta SD dCoP (top) and percentage change in SD dCoP (bottom) indicate a greater sway reduction. EO: Eyes open, EC: Eyes closed; IPT: interpersonal touch.
(TIF)

**S4 Fig** Scatterplot of an individual's balancing skill and interindividual differences in balancing skills. with the benefit of IPT dependent on the personalised performance cluster assignment.
(TIF)

**S5 Fig** Curve fitting results of an individual's balancing skill. With Eyes open (left) and Eyes closed(right) based on age-related motor experience (top) and BMI (bottom).
(TIF)

**S6 Fig** Statistical model of the influential factors on the benefit of IPT. (relative difference in variability in balancing skills with IPT compared to without IPT) for Eyes open condition (top) and Eyes closed condition (bottom).
(TIF)

**S1 Appendix** Results description of a bootstrapped bivariate Pearson correlation analysis between individuals' balancing skill and IPT benefit.
(DOCX)

**S2 Appendix** Results description of serial regression analysis to predict an individual's balancing skill.
(DOCX)

## Author contributions

**Conceptualization:** Katrin Hanna Schulleri, Dongheui Lee, Leif Johannsen.

**Data curation:** Leif Johannsen.

**Formal analysis:** Katrin Hanna Schulleri, Leif Johannsen.

**Funding acquisition:** Dongheui Lee, Leif Johannsen.

**Investigation:** Leif Johannsen.

**Methodology:** Katrin Hanna Schulleri, Dongheui Lee, Leif Johannsen.

**Project administration:** Leif Johannsen.

**Software:** Leif Johannsen.

**Supervision:** Dongheui Lee.

**Writing – original draft:** Katrin Hanna Schulleri, Dongheui Lee, Leif Johannsen.

**Writing – review & editing:** Katrin Hanna Schulleri, Dongheui Lee, Leif Johannsen.

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
