## [Decision Letter · Decision Letter 0]

27 Jan 2025

PONE-D-24-52858The balance stabilising benefit of social touch: influence of an individual’s age and the partner’s relative body characteristicsPLOS ONE

Dear Dr. Johannsen,

Thank you for submitting your manuscript to PLOS ONE. After careful consideration, we feel that it has merit but does not fully meet PLOS ONE’s publication criteria as it currently stands. Therefore, we invite you to submit a revised version of the manuscript that addresses the points raised during the review process.

Please note that we have only been able to secure a single reviewer to assess your manuscript. We are issuing a decision on your manuscript at this point to prevent further delays in the evaluation of your manuscript. Please be aware that the editor who handles your revised manuscript might find it necessary to invite additional reviewers to assess this work once the revised manuscript is submitted. However, we will aim to proceed on the basis of this single review if possible. 

Could you please revise the manuscript to carefully address the concerns raised?

We look forward to receiving your revised manuscript.

Kind regards,

Helen Howard

Staff Editor

PLOS ONE

Journal Requirements:

2. Please describe in your methods section how capacity to provide consent was determined for the participants in this study. Please also state whether your ethics committee or IRB approved this consent procedure. If you did not assess capacity to consent please briefly outline why this was not necessary in this case.

“This work was funded by the Department of Human Movement Science, and the Department of Human-centered Assistive Robotics of TUM, and the DFG SPP 2134 (“The Active Self”) the Deutsche Forschungsgemeinschaft (DFG, German Research Foundation) - 402778716.”

5. Please note that in order to use the direct billing option the corresponding author must be affiliated with the chosen institute. Please either amend your manuscript to change the affiliation or corresponding author, or email us at plosone@plos.org with a request to remove this option.

6. We notice that your supplementary [figures/tables] are included in the manuscript file. Please remove them and upload them with the file type 'Supporting Information'. Please ensure that each Supporting Information file has a legend listed in the manuscript after the references list.

Reviewers' comments:

Reviewer's Responses to Questions

**Comments to the Author**

1. Is the manuscript technically sound, and do the data support the conclusions?

Reviewer #1: Yes

2. Has the statistical analysis been performed appropriately and rigorously? 

Reviewer #1: Yes

3. Have the authors made all data underlying the findings in their manuscript fully available?

Reviewer #1: Yes

4. Is the manuscript presented in an intelligible fashion and written in standard English?

Reviewer #1: Yes

5. Review Comments to the Author

Reviewer #1: The manuscript “The balance stabilising benefit of social touch: influence of an individual’s age and the partner’s relative body characteristics” investigates the relationship between IPT and balance stability. To this aim the authors assessed balance stability and changes due to IPT during single-legged stance, with or without vision, and with or without IPT. Results showed that without vision, the benefit of IPT was determined by the individual's balancing skill and the difference in balancing skill between partners. Vulnerable individuals showed significant improvement with IPT when vision was unavailable. However, with vision, the benefit of IPT was influenced by the individual's age. This study highlights the importance of IPT in improving balance stability, especially for vulnerable individuals when vision is not available.

1) Language – Even if grammatically correct, the general overcomplexity of the sentences risks affecting the manuscript’s readability. It is suggested to adopt a more straightforward style throughout the manuscript, especially in regards to interpolated clauses. The actual style could be a personal choice of the authors, but it risks confusing the reader.

2) Despite providing relevant information, the introduction seems quite long and may overload the reader with too many details. It is suggested to shrink (not cut) it, to better guide readers more straightforwardly through the rationale of the study and towards a focused research question and a clear hypothesis.

3) Single-legged standing sounds quite challenging as an experimental task and may not be representative of everyday balance activities. It could be important to strengthen the rationale for choosing this task. This would avoid risks related to the ecological validity of the findings.

4) The IPT protocol involved participants resting their fingertips against each other. Was the consistency of this contact controlled?

5) To better understand the grouping, it could be helpful to provide more details on the criteria for defining the two clusters and how participants were assigned to these clusters, such as the specific characteristics and thresholds used to differentiate between the clusters.

6) Despite the evident effects in the present study, the influence of IPT goes beyond the behavioral sensorimotor framework. The fit between the present paper and a broader theoretical framework may be broadened. In particular:

- 6.1) At the cognitive level, IPT plays a central role in motor cognition, determining faster responses in body-related mental representation tasks with respect to the touch of objects with similar motor commands, as if targets would be perceived as closer to one’s own body (Martinez et al 2022 Neuroscience). Suggest that IPT enhances the perception of one's own body and its movements, this enhancement could explain why participants in the present study showed improved stability with IPT, as the touch likely facilitated better motor control and awareness of body position.

- 6.2) At the neural level, social distancing and processing of IPT are neurally associated, in that aberrant reactivity of somatosensory regions of the brain (somatosensory cortex and insula) is correlates with ratings of discomfort related to IPT (Maier et al 2020 Am J Psychiatry). The present study supports this by showing that IPT can stabilize balance, potentially by engaging these somatosensory regions in a positive way. This suggests that appropriate IPT can enhance sensory integration and motor responses, reducing discomfort and improving physical stability.

- 6.3) Psychotic conditions leading to increased social distancing such as depression (Bornheimer et al 2022 Soc Work Res), affect the neural responsiveness to IPT (Mielacher et al 2023 Psychol Med). Showing the association between individual interpretation of IPT and social distancing, these studies could complement/integrate the present study’s findings that younger, less skilled individuals benefit more from IPT, especially when paired with a more skilled partner. This highlights the therapeutic potential of IPT in enhancing physical and possibly psychological well-being.

6. PLOS authors have the option to publish the peer review history of their article (what does this mean? ). If published, this will include your full peer review and any attached files.

**Do you want your identity to be public for this peer review?** For information about this choice, including consent withdrawal, please see our Privacy Policy .

Reviewer #1: No

---

## [Author Response · Author response to Decision Letter 1]

18 Apr 2025

Dear Editor and Dear Reviewer,

we like to express our gratitude for the effort you undertook to edit and review our submitted manuscript. We have taken your feedback and suggestions seriously and addressed them in the manuscript accordingly. In the following overview, we have listed each recommendation and our response, and the changes made to the manuscript.

Editorial Comments

1. Response: we have edited our financial disclosure statement in the manuscript and in the cover letter so that the statement is matching in both places.

Response: we have removed all figures from the manuscript and supplementary materials and uploaded them separately.

Response: we taken the best of care to restyle the manuscript according to the published style requirements and we hope that we have not overlooked a detail.

4. Please describe in your methods section how capacity to provide consent was determined for the participants in this study. Please also state whether your ethics committee or IRB approved this consent procedure. If you did not assess capacity to consent please briefly outline why this was not necessary in this case.

Response: we have extended the section about participant recruitment in our manuscript regarding the issue of assessing capacity to consent by inserting a paragraph that details our rationale for assessing the capacity to consent. As the data collection was performed in public a formal assessment of the capacity to consent was not feasible and generally unusual in our behavioural line of research with individuals considered healthy. Instead, a psychologist (L.J.) was evaluated capacity to consent informally through a verbal interaction with interested, potential participants about the purpose of research in human balance control and the presented technical setup. In case that individuals were not able to coherently able to respond to the explanations they were not offered an invitation for participation.

Response: we have edited in the section accordingly, so that the Funding information part in the manuscript and the Financial disclosure statement in the cover letter do match.

Response: we made the correct grant numbers more obvious to the reader and we added a statement that the funders had no role in study design, data collection and analysis, decision to publish, or preparation of the manuscript.

7. Use the direct billing option the corresponding author must be affiliated with the chosen institute. Please either amend your manuscript to change the affiliation or corresponding author.

Response: we do not understand why the primary affiliation of the corresponding author with the RWTH Aachen University as his main affiliation poses a problem here.

8. We notice that your supplementary [figures/tables] are included in the manuscript file. Please remove them and upload them with the file type 'Supporting Information'. Please ensure that each Supporting Information file has a legend listed in the manuscript after the references list.

Response: the supplementary material has been moved into a separate document and a legend of the supporting materials has been included in the manuscript after the reference list.

Reviewer Comments:

9. Language – Even if grammatically correct, the general overcomplexity of the sentences risks affecting the manuscript’s readability. It is suggested to adopt a more straightforward style throughout the manuscript, especially in regards to interpolated clauses. The actual style could be a personal choice of the authors, but it risks confusing the reader.

Response: we agree with the reviewer that in numerous places of the manuscript we used syntactically complex and hard to parse sentence structures. In all sections of the manuscript, we tried hard to identify difficult sentences and simplify these in an appropriate manner.

10. Despite providing relevant information, the introduction seems quite long and may overload the reader with too many details. It is suggested to shrink (not cut) it, to better guide readers more straightforwardly through the rationale of the study and towards a focused research question and a clear hypothesis.

Response: we followed the reviewer’s advice to streamline the introduction both by compressing certain statements and by removing superfluous detail.

11. Single-legged standing sounds quite challenging as an experimental task and may not be representative of everyday balance activities. It could be important to strengthen the rationale for choosing this task. This would avoid risks related to the ecological validity of the findings.

Response: our rationale for asking participants to balance in a single-legged stance was stated in the original manuscript already. However, we moved the specific section to a different location in the manuscript to make it more prominent to the reader. Balancing in single-legged stance is challenging indeed and performance is strongly dependent on age and acquired skill. More importantly, however, it also resembles a posture in which even quite heavy individuals can be disturbed in their balance quite easily. This means that even small individuals with low body mass could exert a perturbing force on a much heavier interaction partner. In a more stable posture such as normal bipedal stance for example, both partners would need to be of more equal mass to effect each other in a biomechanical fashion by the exchange of strong interaction forces (revised manuscript: page 11, paragraph 2).

12. The IPT protocol involved participants resting their fingertips against each other. Was the consistency of this contact controlled?

Response: yes. We constantly checked visually that partners did not grasp each other or did change the initial hand posture. We have clarified this in the methods section of the manuscript. What we were not able to discern was the magnitude of the haptic interaction forces. Here we have the classical conundrum that any measuring device between the two partners would strongly change the haptic interaction in terms of the force feedback but also the mechanics of the tactile contact (revised manuscript: page 12, paragraph 1).

13. To better understand the grouping, it could be helpful to provide more details on the criteria for defining the two clusters and how participants were assigned to these clusters, such as the specific characteristics and thresholds used to differentiate between the clusters.

Response: we realised that our description of the performed cluster analysis might have misled the reviewer. We therefore edited our statement in the methods section and emphasized that we did not a priori constrain the number of clusters to be detected by the algorithm to just two clusters. Two clusters was the result of the cluster analysis without our intervention (revised manuscript: pages 13-14 , paragraph 3).

14. Despite the evident effects in the present study, the influence of IPT goes beyond the behavioral sensorimotor framework. The fit between the present paper and a broader theoretical framework may be broadened.

Response: we thank the reviewer for their insightful suggestions. Yes, we agree that the literature sources hint towards alternative or extended effects of social touch facilitating balance control.

15. At the cognitive level, IPT plays a central role in motor cognition, determining faster responses in body-related mental representation tasks with respect to the touch of objects with similar motor commands, as if targets would be perceived as closer to one’s own body (Martinez et al 2022 Neuroscience). Suggest that IPT enhances the perception of one's own body and its movements, this enhancement could explain why participants in the present study showed improved stability with IPT, as the touch likely facilitated better motor control and awareness of body position.

Response: we added a statement and citation to the discussion section as suggested by the reviewer (revised manuscript: page 25, paragraph 2).

16. At the neural level, social distancing and processing of IPT are neurally associated, in that aberrant reactivity of somatosensory regions of the brain (somatosensory cortex and insula) is correlates with ratings of discomfort related to IPT (Maier et al 2020 Am J Psychiatry). The present study supports this by showing that IPT can stabilize balance, potentially by engaging these somatosensory regions in a positive way. This suggests that appropriate IPT can enhance sensory integration and motor responses, reducing discomfort and improving physical stability.

Response: we added a statement and citation to the discussion section as suggested by the reviewer (revised manuscript: page 25, paragraph 2).

17. Psychotic conditions leading to increased social distancing such as depression (Bornheimer et al 2022 Soc Work Res), affect the neural responsiveness to IPT (Mielacher et al 2023 Psychol Med). Showing the association between individual interpretation of IPT and social distancing, these studies could complement/integrate the present study’s findings that younger, less skilled individuals benefit more from IPT, especially when paired with a more skilled partner. This highlights the therapeutic potential of IPT in enhancing physical and possibly psychological well-being.

Response: we added a statement and citation to the discussion section as suggested by the reviewer (revised manuscript: page 26, paragraph 1).

---

## [Decision Letter · Decision Letter 1]

13 May 2025

The balance stabilising benefit of social touch: influence of an individual’s age and the partner’s relative body characteristics

PONE-D-24-52858R1

Dear Dr. Johannsen,

We’re pleased to inform you that your manuscript has been judged scientifically suitable for publication and will be formally accepted for publication once it meets all outstanding technical requirements.

Kind regards,

Jan Christopher Cwik, Prof. Dr. Dr.

Academic Editor

PLOS ONE

Additional Editor Comments:

Dear authors,

Your manuscript has been reviewed again by the first-round reviewer, who noted that you have incorporated all of his comments regarding the manuscript's revisions to his complete satisfaction. With this in mind, I am pleased to announce that I have decided to accept the manuscript for publication in PLOS One.

Reviewers' comments:

Reviewer's Responses to Questions

**Comments to the Author**

1. If the authors have adequately addressed your comments raised in a previous round of review and you feel that this manuscript is now acceptable for publication, you may indicate that here to bypass the “Comments to the Author” section, enter your conflict of interest statement in the “Confidential to Editor” section, and submit your "Accept" recommendation.

Reviewer #1: All comments have been addressed

2. Is the manuscript technically sound, and do the data support the conclusions?

Reviewer #1: Yes

3. Has the statistical analysis been performed appropriately and rigorously? 

Reviewer #1: Yes

4. Have the authors made all data underlying the findings in their manuscript fully available?

Reviewer #1: Yes

5. Is the manuscript presented in an intelligible fashion and written in standard English?

Reviewer #1: Yes

6. Review Comments to the Author

Reviewer #1: Accept - All my previous comments have been satisfactorily addressed. I recommend to accept the manuscript.

7. PLOS authors have the option to publish the peer review history of their article (what does this mean? ). If published, this will include your full peer review and any attached files.

**Do you want your identity to be public for this peer review?** For information about this choice, including consent withdrawal, please see our Privacy Policy .

Reviewer #1: No

---

## [Editor Report · Acceptance letter]

PONE-D-24-52858R1

PLOS ONE

Dear Dr. Johannsen,

I'm pleased to inform you that your manuscript has been deemed suitable for publication in PLOS ONE. Congratulations! Your manuscript is now being handed over to our production team.

Kind regards,

on behalf of

Prof. Dr. Dr. Jan Christopher Cwik

Academic Editor

PLOS ONE